# Comparison of Glycemic Response to Carbohydrate Meals without or with a Plant-Based Formula of Kidney Bean Extract, White Mulberry Leaf Extract, and Green Coffee Extract in Individuals with Abdominal Obesity

**DOI:** 10.3390/ijerph191912117

**Published:** 2022-09-25

**Authors:** Ewa Lange, Paulina Katarzyna Kęszycka, Ewelina Pałkowska-Goździk, Katarzyna Billing-Marczak

**Affiliations:** 1Department of Dietetics, Institute of Human Nutrition Sciences, Warsaw University of Life Sciences (SGGW-WULS), 159 C Nowoursynowska Street, 02-776 Warsaw, Poland; 2MarMar Investment LLC, ul. Słomińskiego 15/509, 00-195 Warsaw, Poland

**Keywords:** glycemic index, plant extracts, plant-based supplement, postprandial glucose peaks, abdominal obesity

## Abstract

Due to the rising prevalence of obesity and type 2 diabetes, a strategy that can positively influence diet quality in a simple way is being explored, since a low glycemic index (GI) diet is advised in the dietoprophylaxis and diet therapy of diabetes. Methods: Twenty-three women with abdominal obesity participated in the study. The postprandial glycemic response and glycemic index were determined after three carbohydrate meals (noodle soup, white rice, strawberry sorbet) without or with the addition of a plant-base supplement (extracts of kidney bean, white mulberry leaf, and green coffee) with a potentially hypoglycemic effect. For two products (instant noodle soup and white rice), the addition of the plant supplement resulted in a reduction in glicemic iAUC values (respectively, by: 17.1%, *p* = 0.005 and 5.3%; *p* = 0.03; 40.6%, *p* = 0.004 and 5.3%, *p* = 0.019). However, this effect was not observed for strawberry sorbet. The blood glucose concentrations 30 min after the consumption of instant noodle soup and white rice with the plant-based formula addition significantly affected the GI value of tested meals (*p* = 0.0086, r = 0.53; *p* = 0.0096, r = 0.53), which may indicate the effect of this plant supplement on enterohormone and/or insulin secretion. Conclusion: A formula containing kidney bean, white mulberry leaves, and green coffee extracts may therefore be a notable factor in lowering postprandial glycemia and the GI of carbohydrate foods. However, further research is needed to determine for which food groups and meals its use may be most effective.

## 1. Introduction

Obesity and insulin resistance, as a complication of the excessive accumulation of adipose tissue in the body, is still a serious medical problem worldwide despite numerous preventive measures. It is estimated that the prevalence of overweight and obesity has doubled since the 1980s and now covers almost a third of the world’s population [1]. One of the most severe metabolic complications of excessive body weight is diabetes type 2 [2].

The prevalence of glucose abnormalities is increasing medical issue worldwide and it is estimated that almost 500 million people will have prediabetes in 2030 [3] and 629 million (20–79 years), equaling 9.9% of the population, will be living with diabetes in 2045 [4], any measures to change these trends should be implemented.

Obesity, particularly abdominal obesity, is a major risk factor for prediabetes and the progression of prediabetes to diabetes type 2 [5]. Thus, lowering the postprandial glucose levels (‘flattening’ postprandial glycemic response) might be beneficial in metabolic disturbances prevention [6].

The best possible solutions are being searched to not only reach and maintain a patient’s healthy weight but also to increase the effectiveness of glycemic management—the ultimate goal of dietoprophylaxis and diet therapy for diabetes.

Among the dietary recommendations for the prevention and management of diabetes type 2, next to weight reduction, special attention should be paid to diet carbohydrates as the most potent factor influencing postprandial glycemia. Both the quantity and quality of consumed carbohydrates are important, whereby it is preferable to choose products with a low glycemic index [7].

The Glycemic Index (GI) is defined as the area under the 2 h curve of postprandial glucose after the consumption of a food product containing 50 g of digestible carbohydrate and expressed as the ratio of the glycemic response to the same amount of carbohydrate (50 g) from a standard product (glucose or white wheat bread) consumed by the same person [8].

The GI is calculated by comparing the incremental glycemic area under the curve (iAUC) of the test product to the blood glucose response curve of the standard [9]. The higher the glycemic index value of a particular product or meal, the higher the blood glucose concentration after its consumption, in relation to the changes observed after consuming an equivalent amount of glucose [10]. Integrating the GI of the food or meal with the amount of carbohydrates in a given portion size the glycemic load is obtained (GL). The GL provide more accurate picture of the effect of products/meals on postprandial glycemia.

The GI of food products, and therefore glycemic response after consumption, is influenced by a number of factors related to the properties of the product or meal itself, such as the carbohydrate profile (i.e., content and proportions of glucose, fructose, sucrose, lactose). The type and quality of starch is significant as well, including the ratio of amylose to amylopectin, and the production process, which affects the form and structure of the final product, the particle size, and gelification degree of the starch [11]. Postprandial blood glucose levels are also modified by the other food ingredients, including fat, protein, dietary fiber, anti-nutritional components, and organic acids [12,13]. Glycemic response is an individual characteristic; it depends on a variety of factors, including insulin sensitivity, pancreatic secretion, gastrointestinal tract functions and physical activity. The postprandial glycemic response may also be affected by stress, smoking, and medications (e.g., paracetamol) [14].

Postprandial hyperglycemia observed after consuming high-GI foods/meals has been classified as one of the most important risk factors for diabetes type 2 and cardiovascular diseases [15,16,17]. An increased insulin response after a high-GI meal may decrease lipolysis and the utilization of free fatty acids promoting insulin resistance [13]. Results from the Nurses Health Study indicate that high dietary GI is an independent risk factor for ischemic heart disease and myocardial infarction, especially in overweight and obese women, regardless of diet energy [18,19].

Simultaneously, intervention studies indicate that reducing dietary GI values can improve the metabolic control of both type 2 and type 1 diabetes [20]. Reducing dietary GI by an average of 10% corresponds to a reduction in glycosylated hemoglobin in the blood of diabetics, within 7 weeks, by 7.2–8.0%, thereby decreasing the risk of diabetes complications by approximately 10% [21,22]. A lower glycemic and insulin response after eating a low-glycemic-index meal is likely associated with a prolonged feeling of fullness and satiety over time [23]. Low-GI carbohydrate meals may increase fat oxidation compared to meals rich in high GI carbohydrates, in addition to their beneficial effects on postprandial glycemia and insulinemia [24,25]. A meal based on products with a low glycemic index and load may increase postprandial thermogenesis compared to a meal with a high GI and GL [26].

Although the results of studies concerning the effectiveness of diets based on low GI products in obesity management do not allow the formulation of unambiguous conclusions [27], some of these studies indicate positive impact of using low-GI diet plan on weight loss [28,29,30,31].

Some plant extracts are known for containing bioactive ingredients that may inhibit digestion and decrease carbohydrate absorption. Based on a previous study, it was found that preparation consisting of three components—white mulberry leaves extract (*Morus alba L.*), white bean extract (*Phaseolus vulgaris*) and green coffee extract—in chlorogenic acid has the potential to lower the postprandial glucose and insulin level [32].

Thus, the aim of our study was to verify the hypoglycemic effect of dietary supplement consisting of a plant-based formula of kidney bean extract, white mulberry leaf extract and green coffee extract intake before carbohydrate foods in individuals with abdominal obesity.

## 2. Materials and Methods

The study protocol is in accordance with Organization for International Standardization (ISO, 2010) [33] guidelines on glycemic index determination and food classification.

Approval for the study was obtained from the Ethics Committee for Scientific Research with Human Participation at the Faculty of Human Nutrition and Consumer Sciences of the Warsaw University of Life Sciences, decision no: 21_1/2017 of 3.07.2017.

### 2.1. Study Design

To determine the glycemic index of the tested products, the concentrations of glucose in capillary blood were measured: 1) after the consumption of a solution containing 50 g of glucose as a standard and 2) after the consumption of three products (instant noodle soup, white rice, strawberry sorbet) in the amount providing 50 g of digestible carbohydrates alone and followed by a plant-based formula of kidney bean extract, white mulberry leaf extract and green coffee extract in individuals with abdominal obesity. The design of the study is shown in Figure 1.


**The study was conducted in three phases:**


**Phase I.** Determination of two-hour blood glucose levels (by enzyme strip test) after intake of a solution of 50 g glucose—as a standard. The study was carried out in 2 repetitions at a minimum interval of 2 days.

**Phase II.** Determination of two-hour blood glucose level (by enzyme strip test) after consuming the appropriate amount of the tested carbohydrate meal (containing 50 g of digestible carbohydrates). Before the product consumption, study participants were given 150 mL of water to minimize differences between Phase II and Phase III of the study. Meals selected for the study were characterized by: high carbohydrate content (easy to estimate), homogeneity, high availability and acceptability.

During this stage, glucose levels were determined in 2 replicates at minimum 2-day intervals as follows:-after consuming instant noodle soup in an amount that provided 50 g of digestible carbohydrates (75.5 g + 400 mL of water), with 150 mL before meal-after consuming white long-grain rice in an amount that provided 50 g of digestible carbohydrates (65.3 g +124 g water), with 150 mL before meal-after consuming strawberry sorbet in an amount that provided 50 g of digestible carbohydrates (203 g), with 150 mL of water before meal

**Phase III.** Determination of two-hour blood glucose (by strip enzyme assay) after intake of an appropriate amount of a test carbohydrate meal (containing 50 g of digestible carbohydrates), before which participants received 4.5 g of a plant-based formula containing kidney bean, white mulberry leaves, and green coffee extracts in specific proportions dissolved in 150 mL of water. Determination of glucose levels was performed in 2 replicates at intervals of at least 2 days, according to the scheme:-after consuming instant noodle soup in an amount that provided 50 g of digestible carbohydrates (75.5 g + 400 mL of water), with prior intake of a plant-based formula (1 sachet = 4.5 g ± 5%) dissolved in 150 mL of water.-after consuming white long-grain rice in an amount that provided 50 g of digestible carbohydrates (65.3 g + 124 g water), with prior intake of a plant-based formula (1 sachet = 4.5 g ± 5%) dissolved in 150 mL of water.-after consuming strawberry sorbet in an amount that provided 50 g of digestible carbohydrates (203 g), with prior intake of a plant-based formula (1 sachet = 4.5 g ± 5%) dissolved in 150 mL of water.

### 2.2. Tested Plant-Based Formula

The dietary supplement used in this study (commercial name is Tribitor^®^, designed by MarMar Investment LLC, Warsaw, Poland) contained 600 mg of white mulberry extract, 1200 mg of white bean extract and 400 mg of green coffee. White mulberry extract contained 3.12% 1-deoxynojirimycin (standardized to 3% DNJ), white bean extract was standardized to contain a minimum of 4000 α-amylase inhibitor units, and green coffee extract contained 52.3% chlorogenic acid (standardized to a minimum of 50%) and 1.6% caffeine (standardized to a maximum of 2%). The supplement was prepared as a powder dissolved in 150 mL of water at room temperature.

### 2.3. Tested Meals

Three meals characterized by a high carbohydrate content and easy-to-standardize way of preparation were selected for the study. The nutritional values of the tested products were shown in Table 1.

### 2.4. Study Participants

Twenty-five healthy participants were recruited from the volunteer database. Eligible participants were women, 20–55 years old, premenopausal, non-smokers, with abdominal obesity (waist circumference > 80 cm), and without diabetes. The main exclusion criteria were food allergy or intolerance (to reduce the risk of adverse reactions after consuming the test meals or the plant-based formula), metabolic conditions or medications that may affect carbohydrate metabolism.

Qualification for the study was made after medical consultation. Before enrolment in the study, each participant was informed of the purpose, scope and possible risks involved in participation. Anthropometric measurements using standard methods, including waist circumference (with an accuracy of 0.1 cm), height (with an accuracy of 0.1 cm), weight (to the nearest 0.1 kg), and BMI, were calculated for each of the study subjects. Participation in the study was voluntary, and written consent was a mandatory part of inclusion.

Each time, before consuming the study meal, participants fasted for a minimum of 12 h; were instructed not to consume alcohol, large amounts of sweets/carbohydrates and fat; and did not engage in intense exercise.

### 2.5. Methodology for Determining Blood Glucose Level

Capillary blood glucose measurements were made to compare postprandial glucose after consuming carbohydrate meals with or without the addition of 4.5 g of the tested plant-based formula. Glucose determinations in capillary blood were made by the enzymatic method using test strips from Accu-Chek glucometers from Roche. Each time, subjects received a product providing 50 g of digestible carbohydrates and consumed it in no more than 10 min. Further glucose measurements were taken, every 15 min for the first hour and every 30 min for the next hour.

### 2.6. Calculating the GI

To estimate the glycemic index of the meals, a glycemic curve was plotted. It connected the points defining the concentration of glucose in capillary blood at fasting and after the consumption of test meals for the first hour every 15 min, and every 30 min for the next hour. Fasting glucose values were taken as the baseline. The area under the glycemic curve was calculated by summing the areas of the individual trapezoids and triangles and the values were expressed as mg/dL/min. The glycemic index of the test meals was calculated by comparing the incremental area under the curve (iAUC) after consuming the test meals to the area after consuming a standard glucose solution and multiplying the result by 100% [8,33]. The glycemic coefficient of variation GI values, after ingestion of a reference meal, were assumed not to exceed 30% [9]. The effect of the addition the plant-based formula on glycemia and GI values were presented as the percentage of changes in the iAUC and glycemic index values after the test meals with plant formula consumption compared to the iAUC and GI after consumption of the test meals without formula.

### 2.7. Statistical Analysis

Paired tests: one-way repeated measures ANOVA for variables with parametric distribution and Friedman test for variables with non-normal distribution were used to compare glycemia, the areas under the glycemic curve (iAUC) and GI of the tested meals. For the comparison of the changes in the area under the glucose curve and the glycemic index of the test meals after the addition of the plant-based formula analysis of two pairs of dependent variables, the *t*-test for data with a parametric distribution and the rank-signed test for variables with a non-normal distribution were used. Univariate regression analysis based on Spearman’s correlation test was used to evaluate the effect of glycemia on the GI value of meals. The significance level of the analyses performed was taken as *p* ≤ 0.05. Analysis was conducted using Statgraphics Centurion 18.1.12 (Statgraphics Technologies, Inc., The Plains, VA, USA). Unless otherwise indicated, data are presented as mean ± SD, median, standardized skewness, and standardized kurtosis.

## 3. Results

### 3.1. Study Population

The study design included 23 women (two individuals did not meet the inclusion criteria), aged 20–55 years (mean 38.0 ± 9.9 years), premenopausal, with mean body weight 78.9 ± 8.1 kg; BMI 30.0 ± 3.4 kg/m^2^, with abdominal obesity waist circumference 100.8 ± 7.6 cm. The characteristics of the study participants are shown in Table 2.

In the qualification questionnaire, all participants rated their physical activity as low, which was described as: moderate- or vigorous-intensity physical activity less than 150 min of moderate-intensity or 75 min of vigorous-intensity physical activity or the equivalent combination of physical activity a week. All participants were also premenopausal with the regular menstrual cycles (as declared).

### 3.2. Glycemic Response Parameter

The results of fasting blood glucose levels and 15, 30, 45, 60, 90 and 120 min after consumption of the instant noodle soup, rice, and strawberry sorbet (without and followed by consumption of plant-based supplement) are presented in Table 3.

Most participants had normal fasting blood glycemia (60–100 mg/dL); eight subjects had fasting blood glucose values between 100 and 125 mg/dL. Blood glucose levels after carbohydrate meals were within reference limits in all subjects (post-load blood glucose <200 mg/dL, 2 h post-load blood glucose < 140 mg/dL).

The subjects’ maximum blood glucose concentrations were observed 30 min after the consumption of instant soup and cooked rice, but these values did not decrease to the fasting values observed over the next 1.5 h. Similarly, the highest blood glucose concentration was observed at 15–30 min after ingestion of fruit sorbet, but the postprandial glycemia decreased to the values observed in fasting as early as 90 min (Table 3).

The glucose concentrations in given time points after the consumption of glucose (as the standard), tested product and product followed by plant-based supplement are illustrated in Figure 2, Figure 3 and Figure 4.

Comparing the mean blood glucose values after glucose solution ingestion and both soup without or with the plant-based formula, significant differences in blood glucose values were observed at the given time points. Postprandial glycemia after the ingestion of the standard glucose solution were (except for the fasting value, 90 and 120 min time points) significantly higher compared to the values recorded after the consumption of the soup without and with the formula.

When the glycemic levels after eating the soup without and with the preparation were compared, significantly lower glucose concentrations were observed at 30 and 60 min after eating the soup followed by supplement intake (Figure 2).

In comparing the mean glycemic values after ingestion of the glucose solution and rice without or with the addition of the test formula, significant differences were observed at the monitored time points. Blood glucose levels after standard glucose solution ingestion were considerably higher at almost every stage of the study (except for the fasting and at 90 min time points) compared to the values recorded after ingestion of rice alone and with the tested formula. When comparing the postprandial glycemic response after consuming rice and rice with plant-based formula containing kidney bean, white mulberry leaves, and green coffee extracts, significantly lower blood glucose concentrations were observed at 15 min (Figure 3).

When the mean glycemic concentrations after the ingestion of the glucose solution and the sorbet without or followed by the test formula were compared, significant differences in glycemic values were observed. Blood glucose levels after ingestion of the test glucose solution were substantially higher at nearly every stage of the study (except three time points: for the fasting value, at 15 and 120 min) in comparison to the values recorded after ingestion of strawberry sorbet both alone and with the supplement. There were no significant differences in glycemic values after the consumption of sorbet and sorbet with a previously ingested formula at the monitored time points (Figure 4).

### 3.3. Glycemic Index Values

Both the mean and median value of the area under the glycemic curve (iAUC) after the consumption of the standard glucose solution was significantly higher than the iAUC after the consumption of all carbohydrate meals (Table 4). 

In addition, a statistically significantly lower iAUC value was observed for noodle soup, the consumption of which was preceded by the tested formula intake, compared to the iAUC value without plant-based formula solution ingestion (*p* = 0.0169).

The GI values of instant noodle soup with the supplement addition were mainly influenced by changes in glycemia 30 min after its consumption (*p*= 0.0086, r = 0.53), which may indicate the effect of the plant-based formula on insulin secretion after this type of meal (Figure 5).

The GI value of rice with a plant-based formula was mainly influenced by changes in glycemia at 15, 30 and 120 min after the test meal (*p* = 0.0065, r = 0.55; *p*= 0.0096, r = 0.53; *p* = 0.0042, r = 0.57, respectively). Such correlations were not observed for the GI of white rice without the supplement (Figure 6).

The GI values of the fruit sorbet with the plant-based supplement were mainly influenced by changes in glycemia 60 min after consumption (*p =* 0.0376, r = +0.44) (Figure 7).

The GI values of instant noodle soup and white rice consumed after the plant-based supplement were significantly lower than those of meals consumed without the plant-based formula (52.8 vs. 68.3 and 50.8 vs. 64.0, respectively); thus, it can be considered as low GI [34]. In contrast, the addition of the supplement did not considerably affect the GI value of the fruit sorbet (Table 5).

The addition of the plant-based supplement significantly reduced the iAUC and GI after the consumption of soup and white rice (by 17 and 23% and 41 and 23%, respectively) (Table 6).

The glycemic index values of the products varied, which was due to differences in postprandial glycemia. It is worth pointing out that the variation in glycemic response after product consumption by individuals with abdominal obesity may be related to impaired postprandial glucose metabolism and presumably varying degrees of insulin resistance.

## 4. Discussion

The intake of the plant supplement solution before the consumption of instant soup and white rice significantly reduced the postprandial glycemic response, decreased the iAUC, and consequently lowered the glycemic index of these products. A similar effect was not observed for strawberry sorbet.

Postprandial glycemia is influenced by a number of factors in addition to the carbohydrate content, including the content of other food components. The test meals varied in dietary fiber, protein, and fat levels. Maximum blood glucose concentrations in women with abdominal obesity after test meals were lower and were recorded 15–30 min earlier than after a standard glucose solution, although no significant differences in glycemia were observed when a plant-based supplement was added. Increasing the proportion of dietary fiber in a meal may alter postprandial glycemia. Blood glucose concentrations of normal-weight subjects 30 and 45 min after consuming wheat-bran-based cereal snacks, with and without added sugar, were significantly lower than after consuming traditional multi-grain cakes and reached their highest value as early as 15 min after the meal [35]. Similarly, Marangoni and Poli [36] observed a significantly faster reduction in blood glucose 45 min after the consumption of fiber-enriched bread and wheat biscuits compared to blood glucose levels after the consumption of their standard counterparts. Haini et al. [37] also observed a significant reduction in postprandial glucose levels for two hours after consuming a Chinese steamed bun based on amylose-rich maize flour compared to traditional wheat flour products in normal-weight women.

The addition of protein and/or fat to a higher GI carbohydrate meal reduces the glycemic response, probably by gastric emptying and increasing the secretion of insulin and enterohormones that include glucagon-like peptide 1 (GLP-1) and glucose-dependent insulinotropic peptide (GIP) [38,39]. The effect of protein and/or fat on glycemia and postprandial insulinemia may depend on their amount and type, as well as the texture and form of the food [40,41]. The addition of soy protein and corn oil to a liquid carbohydrate test meal reduced the glycemic response, particularly in those with higher waist circumference values [42]. In the present study, the glycemic response and GI after consuming instant noodle soup, which contained the most protein and fat, was not significantly different from the glycemic response after consuming the other carbohydrate meals (starchy-rice and sucrose-rich fruit sorbet) by women with abdominal obesity.

However, we observed that blood glucose concentrations 30 min after the consumption of instant noodle soup and white rice with the plant-based formula addition significantly affected the GI value of tested meals, which may indicate the effect of this supplement on enterohormone and/or insulin secretion.

The supplement analyzed for its effect on the glycemic index contains active components including kidney bean extract, white mulberry leaf extract and green coffee extract, the hypoglycemic effect of which was previously studied. White bean extract contains a mammalian α-amylase inhibitor, which induces structural changes in the active center of the enzyme and may reduce the digestion of starch, maltose, and sucrose, thereby reducing the glycemic response after carbohydrate meals [43,44]. Udani et al. [45] observed that the addition of 3 g of kidney bean extract powder (served with butter as a bread topping) significantly reduced (by approximately one-third) the GI of white bread determined in healthy normal-weight individuals. The average glycemic index of white wheat bread with the addition of white bean extract was between 39 and 45%. In our study, the addition of 4.5 g of the bean extract preparation to the test meals reduced the glycemic index of the tested meals—instant noodle soup (a starchy meal with a higher proportion of fat and protein) and rice (a starchy meal)—and the mean GI values of the test meals were close to 50%.

Yazdankhah et al. [46] observed that the addition of 1.5% white mulberry extract, rich in vitamin C, cyanidin, rutin, and quercetin derivatives, phenolic acids, and tannins, to pasta reduced the in vitro predicted GI value, decreasing the activity of the amylolytic enzymes: α-amylase and α-glucosidase. Mulberry leaf-derived moranolin (marolin-1-deoxynojirimycin) not only inhibits α-glucosidase activity but may also reduce the intestinal glucose uptake by affecting the expression of glucose transporter 2 (GLUT2) and sodium-dependent glucose transporter-1 (SGLT-1) proteins. Its effect on glycemia may also be related to the acceleration of hepatic glucose metabolism by regulating the expression of enzymes involved in glycolysis (glucokinase, phosphofructokinase, pyruvate kinase) and gluconeogenesis (phosphoenolpyruvate carboxykinase, glucose-6-phosphatase) [47]. The white mulberry extract reduced carbohydrate absorption from a mixed meal in healthy adults [48]. The addition of white mulberry leaf extract to a carbohydrate meal also reduced the maximum blood glucose concentration and the value of the area under the glycemic curve in rats, which may indicate that it may be effective in improving glycemic control and improving insulin sensitivity [49,50].

The effect of the plant-based supplement containing white mulberry extract on glucose utilization was indicated by a significant reduction in the area under the postprandial glycemic curve after a carbohydrate meal (rice) and a carbohydrate meal with a higher proportion of protein and fat (instant soup with noodles) in women with abdominal obesity. The glycemic index of the carbohydrate meal (rice) and the carbohydrate meal with a higher proportion of digestible carbohydrates (strawberry sorbet) with the formula containing kidney bean, white mulberry leaves, and green coffee extracts formula was mainly influenced by glycemia at 120 and 60 min after their consumption, which may indicate the effect of the plant supplement on hepatic glycolysis and gluconeogenesis.

Asai et al. [51] noted that the addition of mulberry leaf extract to cooked white rice significantly reduced blood glucose but also insulin concentrations in subjects with impaired glucose tolerance 30 min after ingestion. Similarly, Lown et al. [52] reported that the supplementation of a mulberry leaf preparation with a maltodextrin (high GI) solution significantly reduced both iAUC value of blood glucose and the postprandial insulinemia in subjects with normal body weight and normal glucose tolerance. Mulberry leaf extract also reduced, as determined in healthy adults, the GI of sucrose and maltose, particularly blood glucose concentrations 30 min after a meal [53].

In the present study, the addition of a plant-based supplement did not affect the iAUC and the GI value of sucrose-containing fruit sorbet but reduced the glycemic response after consuming cooked rice, higher in maltose and maltodextrin content, in women with abdominal obesity.

The efficiency of rice starch digestion may be influenced by both the degree of gelification related to the cooking time and the degree of grinding of the grains before cooking [54]. In our study, the glycemia after consuming cooked white uncrushed rice was significantly lower than after consuming a standard glucose solution, but the GI was not significantly different from other carbohydrate meals, although the mean value was within the range of medium GI values (medium-GI foods score 55–70). The addition of supplement with white bean, white mulberry, and green coffee extract significantly reduced both the area under the glycemic curve and the GI of the cooked rice, which can be considered as a low value (GI < 55). Ma et al. [55] observed that the addition of white bean extract to cooked rice (rice porridge) significantly reduced and prolonged starch hydrolysis and the in vitro estimated GI value over time and that this effect was directly proportional to the proportion of this preparation in the meal.

The potential favorable effects on carbohydrate metabolism of the investigated plant-based supplement may be enhanced by the green coffee extract. Green coffee contains phenolic compounds, including chlorogenic acid, with antioxidant activity higher than traditional roasted coffee [56]. In healthy adults, a green coffee extract containing 90–100 mg chlorogenic acid per serving significantly reduced the iAUC value not only compared to a standard glucose solution but also to traditional instant coffees [57]. Eight weeks of regular green coffee consumption by people with type 2 diabetes not only improved fasting blood glucose but also increased the insulin sensitivity and glucagon-like peptide-1 (GLP-1) concentration compared to non-coffee drinkers [58]. Similarly, the beneficial effects of the green coffee extract on fasting blood glucose and insulin sensitivity in individuals with metabolic syndrome were observed by Roshan et al. [59]. Adamska-Patruno et al. [32] recorded that the administration of the plant-based supplement containing kidney beans, white mulberry leaves, and green coffee extracts used in this study reduced the maximum blood glucose and insulin concentrations in normal glycemic subjects at 20–30 min after meals, thereby reducing the occurrence of reactive postprandial hypoglycemia.

Reducing the glycemic index of traditional carbohydrate meals may, by affecting postprandial glycemia and insulinemia and thus modifying the metabolism of adipose tissue, reduce the risk of complications of the metabolic syndrome, especially in women with abdominal obesity [6].

An important limitation of this study is the lack of a normal-weight control group (without abdominal obesity), which makes it impossible to determine the effect of body weight, waist circumference, and BMI on glycemia and GI variations. However, the postprandial blood glucose concentrations after the test meals were referred to the glycemia after ingestion of a standard glucose solution in each individual, allowing for individual variability in the postprandial glycemic response. Interpretation of postprandial blood glucose response, particularly in women with abdominal obesity, would also be facilitated by the assessment of postprandial insulinemia changes, but this would require other methods of obtaining biological material.

## 5. Conclusions

For two out of three meals tested—instant noodle soup and white rice—the addition of the plant-based supplement containing kidney bean, green coffee and white mulberry extracts significantly affected postprandial glycemia in visceral-obesity women and foods’ glycemic index values. Instant noodle soup and white rice consumed after supplement intake had significantly lower glycemic index values than the same products consumed without the formula. Similar relationships were not found for strawberry sorbet.

## Figures and Tables

**Figure 1 ijerph-19-12117-f001:**
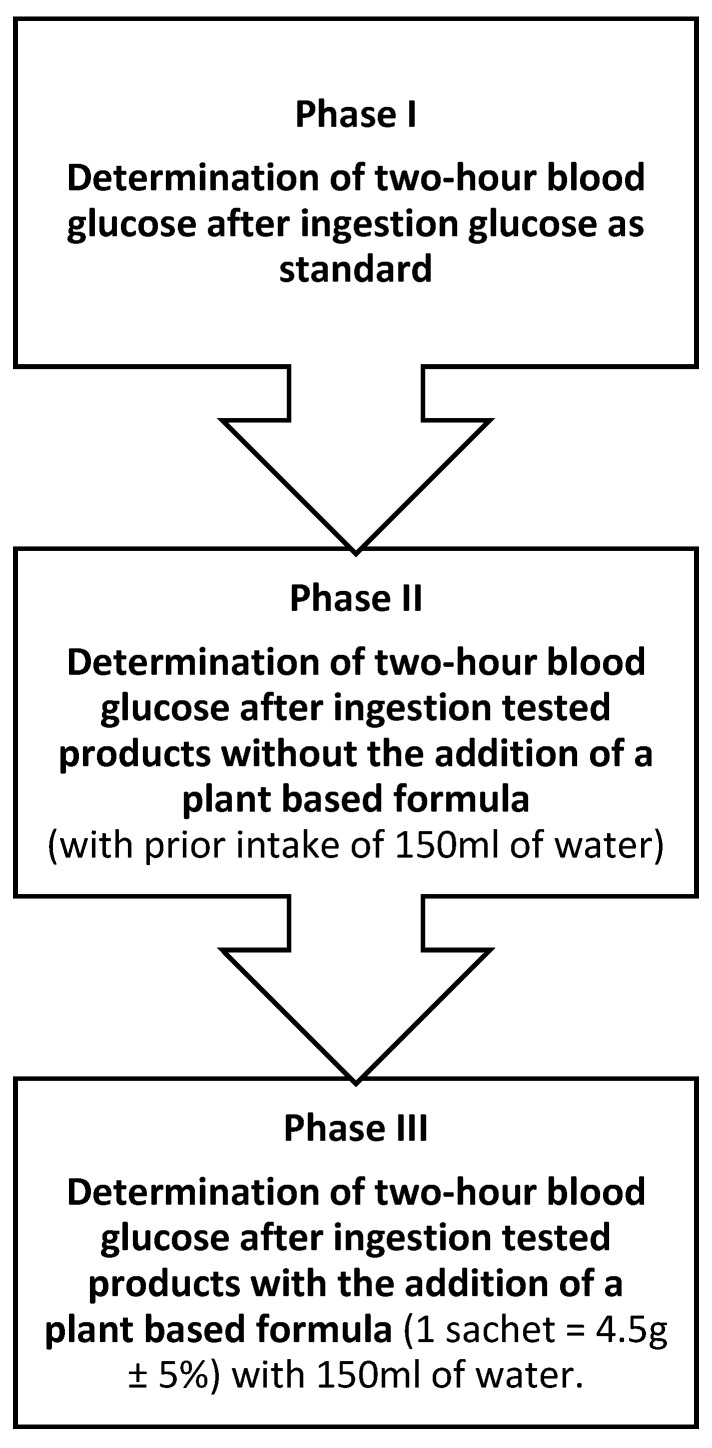
Study design.

**Figure 2 ijerph-19-12117-f002:**
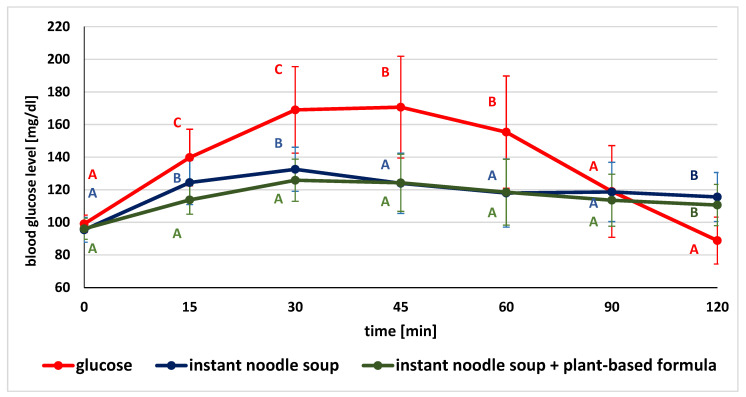
Comparison of mean blood glucose level after consumption of standard glucose solution and instant noodle soup without and with plant-based formula (for variables with the same letter, the difference is not statistically significant; error bars represent standard deviation, *p* > 0.05; one-way repeated-measures ANOVA).

**Figure 3 ijerph-19-12117-f003:**
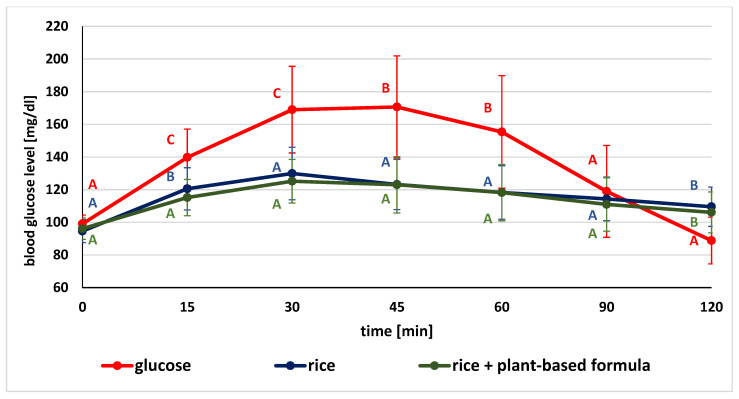
Comparison of mean blood glucose level after consumption of standard glucose solution and white rice without and with plant-based formula (for variables with the same letter, the difference is not statistically significant; error bars represent standard deviation, *p* > 0.05; one-way repeated-measures ANOVA).

**Figure 4 ijerph-19-12117-f004:**
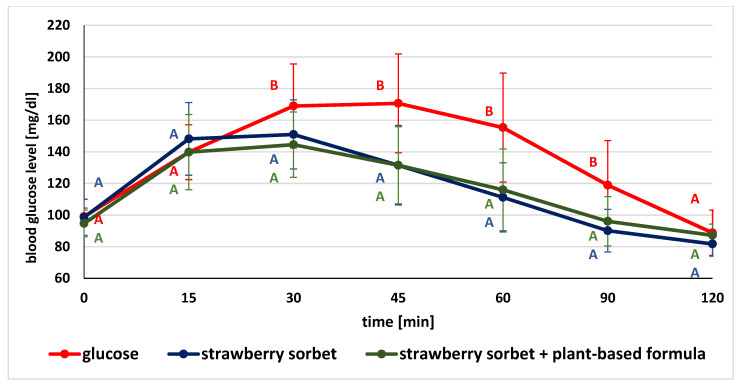
Comparison of mean blood glucose level after consumption of standard glucose solution and sorbet without and with the plant-based formula (for variables with the same letter, the difference is not statistically significant; error bars represent standard deviation, *p* > 0.05; one-way repeated-measures ANOVA).

**Figure 5 ijerph-19-12117-f005:**
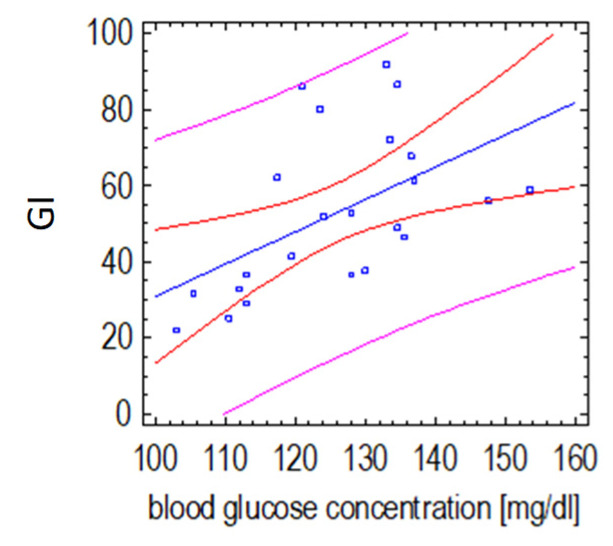
Relationship of the GI value of instant noodle soup with a plant-based supplement and postprandial glucose concentration 30 min after its consumption (Spearman’s correlation test). (the red lines define prediction limits; the pink lines define confidence limits (95%); the blue dots are the results obtained).

**Figure 6 ijerph-19-12117-f006:**
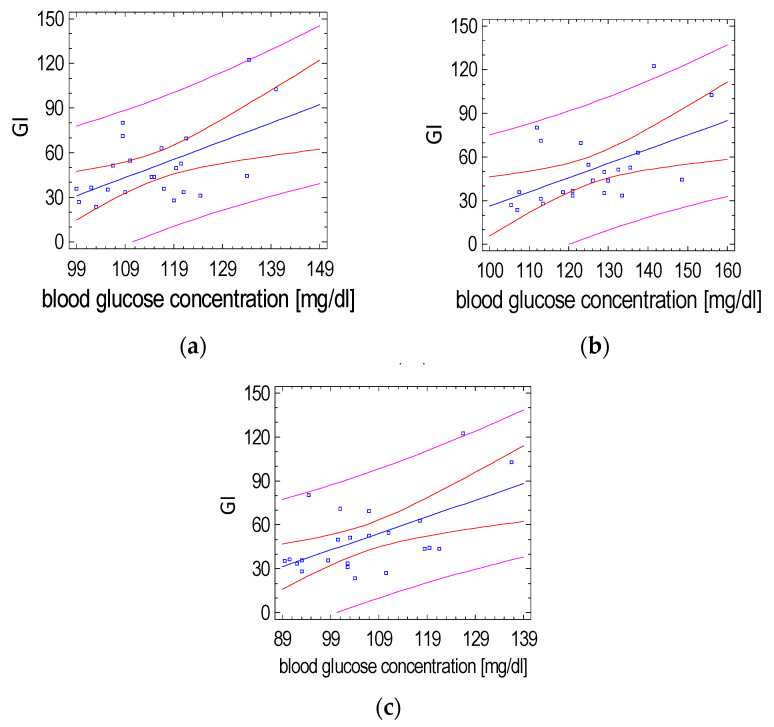
Relationship of the GI value of white rice with the plant-based formula and postprandial glucose concentration at (**a**) 15 min, (**b**) 30 min and (**c**) 120 min after its consumption (Spearman’s correlation test). (the red lines define prediction limits; the pink lines define confidence limits (95%); the blue dots are the results obtained).

**Figure 7 ijerph-19-12117-f007:**
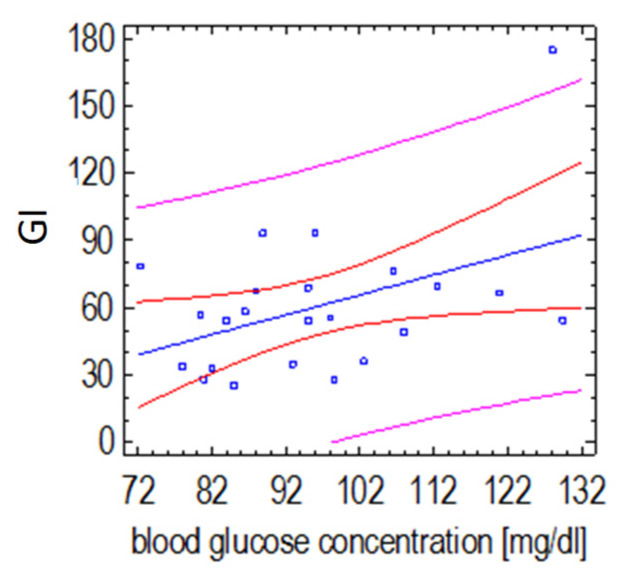
Relationship between the GI value of strawberry sorbet with plant-based formula supplement and postprandial glucose concentration 60 min after its consumption (Spearman’s correlation test). (the red lines define prediction limits; the pink lines define confidence limits (95%); the blue dots are the results obtained).

**Table 1 ijerph-19-12117-t001:** The nutritional value of a tested foods serving providing 50 g of digestible carbohydrates.

Tested Carbohydrate Meals	Quantity of Product (g)	Energy (kcal)	Protein (g)	Fat (g)	Dietary Fiber (g)	Carbohydrate (g)
Instant soup with noodles (before adding the water)	75.5	328	7.9	9.5	2.1	2
White rice (before cooking)	65.3	224	4.4	0.5	1.6	0.1
Strawberry sorbet	203	214	0.8	0.2	1.4	43.7

**Table 2 ijerph-19-12117-t002:** Age and anthropometric parameters of the study population (*n* = 23).

Indicators	Mean * ± SD	Min-Max	Median
Age (years)	38.0 ± 9.9	20–55	42.0
Body weight (kg)	78.9 ± 8.1	67.5–95.5	79.7
Body height (m)	1.62 ± 0.1	1.51–1.74	1.62
Body mass index (kg/m^2^)	30.0 ± 3.4	25.2–37.9	29.3
Waist circumference (cm)	100.8 ± 7.6	90–114	100.0

* normal distribution, skewness and kurtosis not less than −2 and not more than 2; SD—standard deviation.

**Table 3 ijerph-19-12117-t003:** Glycemia after consuming carbohydrate meals providing 50 g of digestible carbohydrates (mean of 2 repetitions) (*n* = 23).

Results	Capillary Blood Glucose Concentration after Consumption of Product without Plant-Based Supplement Intake (mg/dL)							Capillary Blood Glucose Concentration after Consumption of Product with Plant-Based Supplement Intake (mg/dL)						
	Fasting	after 15 min	after 30 min	after 45 min	after 60 min	after 90 min	after 120 min	Fasting	after 15 min	after 30 min	after 45 min	after 60 min	after 90 min	after 120 min
Product:	Instant noodle soup							Instant noodle soup + supplement						
Mean	95.4 ^a(1)^	124.4 ^c^	132.5 ^e^	124.0 ^cd^	118.0 ^b^	118.6 ^bc^	115.5 ^b^	96.1 ^a(1)^	113.8 ^bc^	125.8 ^d^	124.3 ^d^	118.5 ^c^	113.6 ^bc^	110.6 ^b^
SD	7.6	13.6	13.5	18.5	20.9	18.2	15.0	6.4	8.8	12.9	17.6	20.2	16.0	6.4
standardized skewness	0.56	0.20	0.04	0.18	1.09	1.27	1.30	0.14	−0.33	0.12	0.29	1.12	1.11	0.14
standardized kurtosis	−0.51	0.33	−0.55	−0.82	0.54	1.11	1.96	−1.08	0.30	−0.36	−0.37	1.74	1.75	−1.08
Median	95.5	123.0	133.0	116.0	108.0	110.0	110.5	96.0	114.0	128.0	122.0	116.0	112.5	96.0
Product:	White rice							White rice + supplement						
Mean	94.6 ^a^^(^^1)^	120.5 ^cd^	129.9 ^e^	123.2 ^e^	118.3 ^d^	114.3 ^c^	109.5 ^b^	96.1 ^a^^(^^1)^	115.2 ^cd^	125.2 ^e^	123.0 ^de^	118.2 ^cd^	111.0 ^c^	106.1 ^b^
SD	7.1	12.9	16.1	15.4	16.3	13.3	12.0	6.6	11.1	13.3	17.3	17.2	16.4	12.5
standardized skewness	1.42	−0.38	0.08	0.29	0.55	0.33	0.57	0.84	0.58	0.49	0.29	0.46	0.73	0.72
standardized kurtosis	4.79	−0.64	−0.93	−1.32	−0.75	−0.39	0.29	1.60	−0.11	−0.13	−0.61	−0.77	−0.31	0.01
Median	94.0	122.5	130.5	119.0	113.0	114.0	108.0	94.5	115.0	125.0	123.0	114.0	106.5	103.0
Product:	Strawberry sorbet							Strawberry sorbet + supplement						
Mean	98.6	148.2	151.0	131.5	111.2	90.1	81.8	94.7	139.8	144.5	131.5	116.0	96.1	87.2
SD	11.4	23.0	21.9	25.1	21.8	13.5	7.7	8.4	23.7	20.6	24.3	25.8	15.6	7.0
standardized skewness	1.24	0.32	0.96	1.07	1.44	1.39	−0.03	0.25	0.28	0.72	1.03	1.31	0.77	0.41
standardized kurtosis	1.01	0.25	1.90	2.05	3.17	1.92	−0.60	−0.85	−0.68	1.05	1.53	1.21	−0.05	−0.41
Median	96.0 ^b^^(^^2)^	143.5 ^e^	149.5 ^e^	130.5 ^de^	110.5 ^bc^	87.0 ^ab^	80.5 ^a^	93.5 ^ab^	138.5 ^cd^	141.0 ^d^	127.0 ^cd^	108.5 ^bc^	95.0 ^ab^	86.0 ^a^

^(1)^ for variables with the same letter, the difference is not statistically significant; the one-way repeated measures ANOVA. ^(2)^ For variables with the same letter, the difference is not statistically significant; Friedman non-parametric test.

**Table 4 ijerph-19-12117-t004:** The area under the postprandial glycemic curve (iAUC) values (*n* = 23).

	iAUC after Glucose Intake (mg/dL/min)	iAUC after Noodle Soup Consumption (mg/dL/min)	iAUC after Noodle Soup Consumption + Plant-Based Formula (mg/dL/min)
Mean	4641.55 ^C(1)^	2929.98 ^B^	2382.71 ^A^
SD	1873.15	1136.61	1178.14
standardized skewness	0.89	1.10	0.72
standardized kurtosis	−0.25	0.84	−0.13
Median	3920.83	2703.75	2175.00
	iAUC after glucose intake (mg/dL/min)	iAUC after white rice consumption (mg/dL/min)	iAUC after white rice consumption + plant-based formula (mg/dL/min)
Mean	4641.55 ^B(1)^	2697.56 ^A^	2235.78 ^A^
SD	1873.15	1065.61	1052.39
standardized skewness	0.89	−0.25	0.45
standardized kurtosis	−0.25	−0.37	−1.04
Median	3920.83	2722.50	2055.00
	iAUC after glucose intake (mg/dL/min)	iAUC after strawberry sorbet consumption (mg/dL/min)	iAUC after strawberry sorbet consumption + plant-based formula (mg/dL/min)
Mean	4641.55	2346.77	2616.53
SD	1873.15	1110.12	1273.53
standardized skewness	0.89	1.91	1.02
standardized kurtosis	−0.25	5.95	0.64
Median	3920.83 ^B(2)^	2177.60 ^A^	2457.90 ^A^

^(1)^ for variables with the same letter, the difference is not statistically significant *p* > 0.05; the one-way repeated measures ANOVA (respectively, for: noodle soup *p* = 0.0169 and white rice *p* = 0.0126); ^(2)^ for variables with the same letter, the difference is not statistically significant *p* > 0.05. Friedman non-parametric test (*p* > 0.0001).

**Table 5 ijerph-19-12117-t005:** Glycemic index of the tested products without and with the addition of the plant-based formula (*n* = 23).

	IG of Noodle Soup	IG Noodle Soup + Supplement	*p* Value
Mean	68.34	52.84	NI
SD	28.98	20.66	
standardized skewness	0.56	0.40	
standardized kurtosis	4.58	−0.85	
Median	64.06 ^B(1)^	51.89 ^A^	0.0049
	**GI of white rice**	**GI of white rice** **+ supplement**	
Mean	64.03 ^B(2)^	50.81 ^A^	0.0189
SD	30.18	24.73	
standardized skewness	0.51	1.54	
standardized kurtosis	0.48	2.32	
Median	62.13	43.67	NI
	**GI strawberry sorbet**	**GI strawberry sorbet + supplement**	
Mean	53.36	60.48	NI
SD	20.52	31.88	
standardized skewness	0.56	2.14	
standardized kurtosis	0.34	6.89	
Median	55.45 ^A(1)^	55.48 ^A^	0.3229

^(1)^ paired signed rank test; ^(2)^ paired sample *t*-test; NI—not indicated.

**Table 6 ijerph-19-12117-t006:** The effect of the addition of the plant-based formula on glycemia and GI values ^(^^1)^.

	The Impact of Supplement on iAUC—Noodle Soup (% of Changes)	The Impact of Supplement on iAUC—White Rice (% of Changes)	The Impact of Supplement on iAUC—Strawberry Sorbet (% of Changes)
*n*	23	23	23
Mean	−17.06 ^(2)^	−5.28 ^(2)^	18.84 ^(^^2)^
SD	32.34	59.41	47.07
standardized skewness	0.02	−0.79	−1.06
standardized kurtosis	0.67	−0.11	0.81
Median	−25.22	−22.66	4.87
*p* value	0.0053	0.02915	0.20001
	**The impact of supplement on GI—noodle soup (% of changes)**	**The impact of supplement on GI—white rice (% of changes)**	**The impact of supplement on GI—strawberry sorbet (% of changes)**
*n*	23	23	23
Mean	−40.65 ^(2)^	−5.28 ^(2)^	5.37 ^(2)^
SD	60.80	59.41	42.45
standardized skewness	1.21	0.27	−1.25
standardized kurtosis	0.63	−0.72	0.78
Median	−33.73	−22.66	−8.02
*p* value	0.0039	0.0189	0.1825

^(1)^ Changes in the iAUC and glycemic index after consumption of the test meals with plant-based formula compared to the iAUC and GI after consumption the test meals without formula (presented as a percentage); ^(2)^ paired sample *t*-test.

## Data Availability

Detailed data are available from the authors of the study, if necessary, please contact us.

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
