# Peer review of "Comparison of Glycemic Response to Carbohydrate Meals without or with a Plant-Based Formula of Kidney Bean Extract, White Mulberry Leaf Extract, and Green Coffee Extract in Individuals with Abdominal Obesity"

_ijerph, 2022, doi:10.3390/ijerph191912117_

Round 1
Reviewer 1 Report (Previous Reviewer 2)
The authors have made the corrections and answered the questions satisfactorily. Therefore I recommend the acceptance of the manuscript
Author Response
Thank you very much for the answer and acceptance of our explanations and changes.
Reviewer 2 Report (New Reviewer)
1. Why are only women recruited? Is there any gender difference?
2. Why is the recruitment age only 20-55 years old? What is the reference?
3. Is there any comparison between the daily physical activity levels of different groups at baseline? This may affect Glycemic Response.
4. The study needs to provide a flow chart.
Author Response
Thank you very much for your comments and comments.
Of course, we will adapt the submitted manuscript accordingly.
The Reviewer’s comment: Why are only women recruited? Is there any gender difference?
Author’s Response: Despite no indication in the ISO guidelines [1] regarding the age or the gender of the study participants, we decided to focus on women to preserve the group' homogeneity and to eliminate possible gender-dependent differences in glucose metabolism [2, 3, 4]. It would also be interesting to carry out the study according to the presented schedule in the male population and compare the results.
- International Organization for Standardization. ISO 26642: 2010: Food Products - Determination of the Glycaemic Index (GI) and Recommendation for Food Classification; ISO: Geneva, Switzerland, 2010.
- Blaak E. Sex differences in the control of glucose homeostasis. Curr Opin Clin Nutr Metab Care. 2008, 11(4):500-4. doi: 10.1097/MCO.0b013e32830467d3.
- Varlamov O, Bethea CL, Roberts CT Jr. Sex-specific differences in lipid and glucose metabolism. Front Endocrinol (Lausanne). 2015, 5:241. doi: 10.3389/fendo.2014.00241.
- Mauvais-Jarvis F. Gender differences in glucose homeostasis and diabetes. Physiol Behav. 2018, 187:20-23. doi: 10.1016/j.physbeh.2017.08.016.
The Reviewer’s comment: Why is the recruitment age only 20-55 years old? What is the reference?
Author’s Response: Thank you for this point. There is evidence that glucose metabolism changes with age [1,2]. The risk of developing metabolic syndrome increases with age - according to Kuku et al. (2010) metabolic syndrome was present in 26% of participants aged ≤65 years vs. 55.0% of those over 65 years [3]. Older age (> 65 years old) is also a risk factor for diabetes [4]. With this in mind, we considered the upper age limit of the female participants to be 55 years, moreover each of them declared regular menstruation cycles, so we could conclude that we had premenopausal women in these age ranges.
- Chia CW, Egan JM, Ferrucci L. Age-Related Changes in Glucose Metabolism, Hyperglycemia, and Cardiovascular Risk. Circ Res. 2018, 123(7):886-904. doi: 10.1161/CIRCRESAHA.118.312806.
- Basu R, Dalla Man C, Campioni M, Basu A, Klee G, Toffolo G, Cobelli C, Rizza RA. Effects of age and sex on postprandial glucose metabolism: differences in glucose turnover, insulin secretion, insulin action, and hepatic insulin extraction. 2006, 55(7):2001-14. doi: 10.2337/db05-1692.
- Kuk JL, Ardern CI. Age and sex differences in the clustering of metabolic syndrome factors: association with mortality risk. Diabetes Care. 2010, 33(11):2457-61. doi: 10.2337/dc10-0942.
- Bellary S, Kyrou I, Brown JE, Bailey CJ. Type 2 diabetes mellitus in older adults: clinical considerations and management. Nat Rev Endocrinol. 2021, 17(9):534-548. doi: 10.1038/s41574-021-00512-2.
The Reviewer’s comment: Is there any comparison between the daily physical activity levels of different groups at baseline? This may affect Glycemic Response.
Author’s Response: Thank you for pointing this out. In the qualification questionnaire, we had a question regarding the subjective assessment of the physical activity level. All participants rated their physical activity as low, which was described as: moderate- or vigorous-intensity physical activity less than 150 minutes of physical activity a week or 75 minutes of vigorous-intensity physical activity or the equivalent combination. Given that all participants made the same indications, we could conclude that the level of physical activity was not a factor significantly modifying/ differentiating the results.
https://health.gov/our-work/nutrition-physical-activity/physical-activity-guidelines/current-guidelines
The Reviewer’s comment: The study needs to provide a flow chart.
Author’s Response: As it was recommended, in revison, the study flowchart has been added.

This manuscript is a resubmission of an earlier submission. The following is a list of the peer review reports and author responses from that submission.
Round 1
Reviewer 1 Report
The aim of our study was to verify the hypoglycaemic effect of a plant-based formula of Kidney bean extract, white mulberry leaf extract and green coffee extract intake before three different foods with the same carbohydrate content in individuals with abdominal obesity.
The first indication that I must make is that I am not sure if the study corresponds to the objectives of IJERPH. In the introduction they try to relate it to the prevention of obesity at the population level.
Noodle soup, white rice and strawberry sorbet do not seem like everyday foods for the general population. They seem exotic to me and I don't see their use in avoiding the diabetes pandemic we are suffering from.
On the other hand, it is not clear to me why extracts of Kidney bean, white mulberry leaf, and green coffee are used together. The interesting thing would have been to see the effectiveness of each complement separately. This study appears to be to assess the efficacy of Tribitor®. Thus, it seems that the interest is more commercial than scientific. KBM was an employee of MarMar Investment LLC and one of the inventors of the product.
Capillary blood glucose is not a very accurate method for determining plasma glucose.
The participants claim to be healthy and without diabetes, but we do not know if metabolic syndrome has been ruled out (eight subjects had fasting blood glucose values ​​between 100 and 125 mg/dl).
I think they should have been better diabetes patients.
On the other hand, the sample size is not calculated.
In view of figure 3, it seems that the differences between taking Tribitor or not taking it may be statistically significant but not clinically evident.
Reviewer 2 Report
After careful consideration, I fell that the manuscript entitled “Comparison of Glycemic Response to Carbohydrate Meals Without or with A Plant-Based Formula of Kidney Bean Extract, White Mulberry Leaf Extract, and Green Coffee Extract in Individuals with Abdominal Obesity” has merit, but is not suitable for publication as it currently stands. Therefore, my decision is "Major Revision."
In general, I believe that some statistical analyzes are not correct. The main point is that all analyzes performed must consider paired tests (one-way repeated measures anova instead of (standard) anova, friedman test instead of kruskall-wallis test, etc...). This is because all tests are performed on measurements of the same individuals (all tests consider the group of 23 women)
Here are my comments:
ABSTRACT
- lines 18-19: to adopt "." to separate decimals
STATISTICAL ANALYSIS
- There was no description of which post-hoc was adopted in non-paired tests (comparing three or more groups);
- The authors describe that a regression analysis based on Spearman's correlation was performed. However, no (uni and multivariate) regression analysis was performed in the work.
- Descriptions of other tests must be included (see comments below)
RESULTS
- Table 3: The statistical analysis performed is not correct. The authors describe having performed an ANOVA (witch LSD post-hoc test) and the Kruskall-Wallis test, however the correct use is tests for paired data (one-way repeated measures anova and Friedman's non-parametric test). Furthermore, these tests must be described in the statistical analysis section.
- Figures 1-3: The letters and error bars in the figures could be of different colors so that they can be identified from which group they represent. Furthermore, the authors must indicate what the points in the figures represent (mean or median). It is also necessary to indicate what the error bars represent (standard deviation, standard error or confidence intervals?). In addition, the correct statistical test is the one-way repeated measures ANOVA and Friedman non-parametric test.
- Table 4: In Section 2.6, lines 189-193: the authors describe how the iAUC is obtained. In this description it is understood that the iAUC ranges between 0% and 100%. However, all results of iAUC presented in Table 4 are greater than 2000 and are also not in percentage. Check for this inconsistency. In addition, the correct statistical test is the one way repeated measures ANOVA and Friedman non-parametric test.
- Tables 5 and 6: The signed rank test and paired t-test were not described in Statistical Analysis section. Why did the authors choose the sign test instead of the Wilcoxon rank test?